# An Integrative Migraine Polygenic Risk Score Is Associated with Age at Onset But Not Chronification

**DOI:** 10.3390/jcm13216483

**Published:** 2024-10-29

**Authors:** Bruce A. Chase, Roberta Frigerio, Susan Rubin, Tiffani Franada, Irene Semenov, Steven Meyers, Stuart Bergman-Bock, Angela Mark, Thomas Freedom, Revital Marcus, Rima Dafer, Jun Wei, Siqun L. Zheng, Jianfeng Xu, Ashley J. Mulford, Alan R. Sanders, Anna Pham, Alexander Epshteyn, Demetrius Maraganore, Katerina Markopoulou

**Affiliations:** 1Department of Health Information Technology, Endeavor Health, Skokie, IL 60077, USA; 2Pritzker School of Medicine, Chicago, IL 60637, USA; 3Research Institute, Endeavor Health, Evanston, IL 60201, USA; 4Department of Neurology, Endeavor Health, Evanston, IL 60201, USA; 5University of Chicago Pritzker School of Medicine, Chicago, IL 60637, USA; 6Department of Neurological Sciences, Rush University Medical Center, Chicago, IL 60612, USA; 7Center for Individualized Medicine, Endeavor Health, Evanston, IL 60201, USA; 8Genomic Health Initiative, Endeavor Health, Evanston, IL 60201, USA; 9Department of Psychiatry and Behavioral Neuroscience, University of Chicago, Chicago, IL 60637, USA; 10Department of Neurology, Tulane University, New Orleans, LA 70112, USA

**Keywords:** migraine, polygenic risk score, electronic health record (EHR) review, real-world study, structured clinical documentation tools

## Abstract

**Background/Objective**: Genome-wide association studies (GWASs) demonstrate a complex genetic landscape for migraine risk. Migraine polygenic risk scores (PRSs) developed from GWAS data may have utility for predicting disease course. We analyzed the strength of association between an integrative migraine PRS and age at onset and chronification. **Methods**: In this retrospective clinical/genetic case–control study, PGS004799 was calculated for adults with European ancestry from two real-world community cohorts. In the DodoNA cohort, 1653 treated, deeply phenotyped migraine cases, diagnosed using International Classification of Headache Disorders 3rd edition criteria, were followed for a mean (range) of 2.3 (1–10) years and compared to 3460 controls (without migraine diagnosis). In the GHI cohort, 2443 cases were identified using the first migraine ICD code as a proxy for migraine onset and compared to 8576 controls (without migraine ICD codes). PRS associations with age at onset (DodoNA) or first migraine ICD code (GHI) and chronification (DodoNA) were evaluated. **Results**: In both cohorts, PRS was higher in cases (DodoNA mean (range) cases: 0.82 (0.07–1.76), controls: 0.78 (0.04–1.56); *t* (5111) = −6.1, *p* = 1.4 × 10^−9^, GHI: cases: 0.79 (0.003–1.68), controls: 0.75 (−0.06–1.53); *t* (11,017) = −7.69, *p* = 1.6 × 10^−14^), and a higher PRS was associated with earlier onset in females (HR [95% CI] DodoNA: 2.1 [1.6–2.6, *p* < 0.001; GHI: 1.8 [1.4–2.1], *p* < 0.001) and in males (DodoNA: 2.5 [1.3–4.7], *p* = 0.005; GHI: 1.6 [1.1–2.6], *p* = 0.027). PRS was not different in cases with or without chronification (t (1651) = −1.67, *p* = 0.094) and was not associated with earlier chronification (1.2 [0.8–1.6], *p* = 0.424). **Conclusions**: Higher genetic risk was associated with earlier onset and increased risk of migraine well into adulthood, but not with chronification. This suggests that the PRS quantifies genetic susceptibility that is distinct from factors influencing disease course.

## 1. Introduction

Migraine is a common neurological disorder, with a worldwide prevalence of 14.4% (females: 13.8–15%; males: 9.4–10.2%) [1]. Diagnostic criteria are outlined in the International Classification of Headache Disorders 3rd edition (ICHD-3) [2,3,4]. Migraine can be classified as episodic or chronic and by the presence or absence of aura [3]; however, the variable clinical manifestations of migraine have complicated attempts to classify patients using migraine-associated characteristics and subtypes [5,6,7,8,9]. Furthermore, migraine is a complex genetic disorder associated with both phenotypic heterogeneity and familial aggregation, features typically associated with polygenic inheritance.

Genome-wide association studies (GWASs) using case/control cohorts have demonstrated the complex genetic landscape for migraine risk and provided insights into migraine etiology, familial aggregation [10,11,12,13,14,15,16,17,18,19,20,21,22,23], and underlying biological pathways. Migraine is associated with genes involved in arterial and smooth muscle function, neuronal function/development, nociception, glutamate and calcium homeostasis, oxidative stress, calcitonin gene-related peptide pathways, nitric oxide and serotonin signaling, and intestinal and musculoskeletal/connective tissue function. Some variants show specific associations with sex, migraine with aura, or migraine without aura [18,19,20,21,22,23].

From both clinical and research perspectives, it is useful to understand the strength of association between genetic factors and age of migraine onset or disease outcomes such as chronification [24,25,26,27]. As individual migraine-risk variants have small effect sizes, composite scores are used to describe overall genetic risk. A polygenic risk score (PRS) captures risk at independent loci with genome-side significance and potentially non-independent loci with smaller effect sizes [28]. Though single risk variants or high risk scores do not automatically effect disease course, severity, or treatment response [29,30,31], PRSs can identify risk classes that complement other types of risk prediction, and so may have clinical or personalized utility [24,32,33,34]. Common polygenic variation captured by PRSs was shown to contribute to the familial aggregation of migraine [16], with mean PRS being higher in age groups with onset under age 20 [16], and, in a questionnaire-based study, PRSs correlated with ICHD-3 diagnostic criteria for migraine [10]. However, a previously developed PRS failed to show an association with chronic migraine [35].

In this report, we build upon these efforts and evaluate a recently developed PRS with improved prediction accuracy in treated patients from two real-world, community cohorts, one of which was deeply phenotyped. We assessed the strength of association between migraine genetic risk and age at onset in both cohorts and re-examined the association between migraine genetic risk and chronification over the disease course in the deeply phenotyped cohort.

## 2. Materials and Methods

### 2.1. Study Patients, Migraine Diagnostic Criteria, and Data Elements

In this retrospective clinical–genetic study, we evaluated PRS associations in patients from two community-based cohorts. Inclusion criteria required all participants to be aged 18 years or older and a resident of Northeastern Illinois, USA. One cohort (DodoNA; N = 8515) was a subset of *The DodoNA Project: DNA Predictions to Improve Neurological Health*, which investigates the contribution of genetic risk to the progression and outcomes of neurological diseases. Cases were diagnosed using ICHD-3 criteria and deeply phenotyped through extensive clinical interviews with neurologists who used a migraine structured clinical documentation (SCDS) toolkit embedded in the electronic health record (EHR) at the initial visit (baseline) and longitudinally at annual follow-up visits [36,37]. The toolkit (described in Appendix A) provided a vehicle to standardize the collection of clinical data across neurologists and patients over time, so that over 200 discrete data elements, including assessments of validated measures of migraine disability, could be efficiently collected during a routine clinical encounter. A second cohort was obtained from *the Genomic Health Initiative* (GHI) project (N = 16,776), which enrolls patients from all clinical areas (Table 1). The GHI cohort was expected to have the migraine incidence of an unselected clinical population.

All patients provided written informed consent for genotyping and EHR review. Studies were approved by the NorthShore University HealthSystem Institutional Review Board (DodoNA, EH10-139, April 2011; GHI, EH14-092, February 2014). Data were collected from study approval through June 2023. A flow diagram of patient enrollment in each cohort is presented in Figure 1.

Participants were excluded (DodoNA, N = 1656; GHI, N = 2941) if they had non-European (EUR) ancestry, because the PRS was developed from, and evaluated on, GWASs with EUR ancestry. As both cohorts had a low percentage of non-EUR ancestry, we could not reliably evaluate associations with PRS developed for other populations. DodoNA patients with migraine but not followed with the migraine toolkit, and DodoNA and GHI patients with non-migraine headache ICD codes were also excluded (DodoNA N = 1746; GHI N = 1595). Patients enrolled in both studies (N = 1221) were included only in the DodoNA migraine cohort.

The DodoNA cohort included 1653 cases: treated patients living with migraine, drawn from a real-world, community-based practice setting. Controls were other DodoNA participants (N = 3460) without a migraine or headache disorder (i.e., ICD codes: ICD9 346-, 339-, 307.81, 784.0; ICD10 G43-, G44-). The GHI cohort included 2443 cases (diagnosis based on ICD code in the EHR) and 8576 controls without the migraine or headache ICD codes detailed above.

Case–control studies evaluated risk-score associations with age at first migraine onset in the DodoNA cohort or at first ICD code diagnosis (ICD9 code 346- or ICD10 code G43-) as a proxy for first migraine diagnosis in the GHI cohort. Risk-score associations with chronification (≥15 migraine days per month for at least three months) were evaluated in 709 cases in the DodoNA cohort that had documented chronification of migraine during their disease course. The year of chronification onset was recorded at the initial visit, if it occurred prior to study enrollment, or at a subsequent visit if it occurred during follow-up.

### 2.2. Genotyping and Migraine PRS Calculation

For the DodoNA cohort, single-nucleotide polymorphism (SNP) genotyping on an Affymetrix Axiom^TM^ array with custom content and quality control measures were as described previously [32]. The TOPMed imputation server was used for genotype imputation, phasing, and ancestry estimation [38,39,40].

For the GHI cohort, 0.5× -coverage whole-genome sequencing was obtained using Illumina NovaSeq 6000 (GeneByGene, Houston, TX, USA). After alignment, quality control, and imputation using the 1000-Genomes Phase-3 reference panel with *loimpute* (Gencove, Long Island City, NY, USA), variant call files were checked for sex concordance. Ancestry was determined using principal component analysis with ancestry-informative markers.

For each participant, we calculated the integrative migraine PRS PGS004799, which was recently developed using the PRSmixPlus pipeline by Truong et al. [40,41]. The PRSmixPlus pipeline integrates multiple PRSs for a target population, which leads to improved PRS prediction accuracy. PGS004799 was originally developed and evaluated using 37,851 and 9462 individuals, respectively, with EUR ancestry from the All of Us cohort. In the DodoNA cohort, PGS004799 was calculated using 2938,299 SNPs that had a minor allele frequency > 0.005, *r*^2^ > 0.8, and that were also present in the Phase 3 1000-Genome reference panel. In the GHI cohort, PGS004799 was calculated using 2917,999 SNPs with a genotype probability ≥ 0.9.

Risk-score mean, standard deviation (SD), median, and range are reported for patient groups. Mean differences were assessed using a two-tailed *t*-test. PRS distributions were visualized using a kernel-density plot obtained using an Epanechnikov kernel function. A Kolmogorov–Smirnov test evaluated the equality of PRS distribution between cases and controls (DodoNA, GHI) and between cases with and without chronification (DodoNA). This is the primary analysis of these data. Statistical analyses were performed using Stata BE 18.0.

### 2.3. PRS Associations with Migraine Age at Onset and Chronification

Cox proportional hazards regression evaluated PRS associations with age at onset (DodoNA) or at first migraine ICD code (GHI). When estimates of baseline hazards differed by sex (see the Results section), separate models were developed for each sex. Genetic principal components were included as covariates in each model. Power calculations used the available numbers of study participants, the observed PRS standard deviation (0.22), a probability of event or censoring of 0.50, and an *R*^2^ of 0.05 for cases regressed on PRS, genetic principal components, and age. The Cox proportional hazard regression analyses had 80% power at α = 0.05 to detect hazard ratios (HRs) of 1.39 (females) and 2.10 (males) for age of onset in the DodoNA cohort, HRs of 1.24 (females) and 1.33 (males) for age at first migraine ICD code in the GHI cohort, and an HR of 1.57 for chronification in migraine cases in the DodoNA cohort.

In the DodoNA cohort, cases were censored at neurologist-documented, patient-reported age at onset. Controls were censored at the last known clinical encounter. We also assessed the relationship between first migraine ICD code and patient-reported age at onset in patients grouped by onset relative to 2003 in the DodoNA cohort, since our EHR began in 2003. In the GHI cohort, cases were censored at the age of first migraine ICD code and controls at the last known clinical encounter. Since patient-reported onset age may have been more reliable in younger patients, test–retest reliability was unknown, and first migraine ICD code may not have been a reliable estimate of onset age, additional sensitivity analyses were performed using age quartiles instead of patient-reported onset age (DodoNA) or first migraine ICD code age (GHI) as the analysis time.

Three related analyses evaluated risk-score associations with age or disease duration at chronification in cases in the DodoNA cohort. In each, cases were censored at chronification, as determined by the treating neurologist. In the first analysis, controls were patients without migraine and migraine patients without chronification. In the second and third analyses, controls were migraine patients without chronification. Analysis time was age in the first and second analyses and disease duration (years from onset) in the third. Similar results were obtained in sensitivity analyses using quartiles of analysis time.

Kaplan–Meier analysis assessed whether survival free of migraine diagnosis, migraine ICD code, and/or chronification of migraine varied across groups defined by risk-score tertiles. Between-group differences were considered significant if Bonferroni-adjusted *p* < 0.05.

The goodness of fit of the Cox proportional hazard models were assessed visually by plotting the estimated cumulative hazard function for the Cox–Snell residuals relative to the residuals themselves. This demonstrated that, except at long analyses times, all models satisfied the proportional hazards assumption. All models also passed a global test of the proportional hazards assumption based on the Schoenfeld residuals. The proportional hazards assumption in Kaplan–Meier analyses was assessed by examining whether the curves generated by plotting ln (analysis time) versus –ln{-ln (survival)} for risk-score tertiles were parallel. This was the case for all analyses where between-group differences were significant. Appendix A presents the model-fit data.

## 3. Results

### 3.1. Study Population Demographics

Cases in the DodoNA cohort were well-characterized patients living with migraine disease who were treated by neurologists in a headache clinic. They had a wide range of clinical and demographic characteristics, varying in age, disease duration, onset age, and objective assessments of migraine disability (Table 1). Cases in the DodoNA cohort had relatively high incidence of chronification (N = 709, 42.9%). This reflects their treatment within a headache clinic, recruitment by neurologists into a migraine study, as well as the non-cross-sectional assessment of chronification over the duration of migraine disease, including follow-up for a mean (range) of 2.3 (1–10) years. Cases in the GHI cohort, and controls in both the DodoNA and the GHI cohort, were identified based on ICD code annotation in the EHR, which is similar to how cases and controls are defined in GWASs.

### 3.2. PRS Distribution Skews Higher in Patients with Migraine

In both cohorts, the mean and median PRS was higher in cases than in controls, and the PRS distribution was different between cases and controls (Table 2, Figure 2A,B). The overlap in the distribution of PRSs between cases and controls is consistent with previous assessments of genetic risk scores for prevalent conditions like migraine, where not all cases seek treatment and/or where genetic risk confers susceptibility but non-genetic factors trigger onset [39,40]. Within DodoNA cases, the PRS did not differ between patients with and without migraine chronification (Figure 2C).

### 3.3. Higher Risk Scores Are Associated with Earlier Migraine Onset

When baseline hazard estimates differed by sex (Figure 3A, Figure 4A and Figure 5A), male and female cases were analyzed separately. A higher PRS was associated with earlier migraine onset in the DodoNA cohort (Figure 3C–H) and with earlier age at first migraine ICD code entry in the GHI cohort (Figure 4B–G), whether analysis time was age or age quartile. Consistent with these findings, in the DodoNA cohort, females differed in onset age by PRS-score tertile, with higher tertiles (higher PRSs) having earlier onset (Figure 3G,H). In the GHI cohort, females with third (highest)-tertile PRS scores had earlier age at first migraine ICD code entry than females with first- and second-tertile scores (Figure 4F,G). In the DodoNA migraine cohort, males in different PRS-score tertile groups did not differ in onset age (Figure 3E,F), while males in the GHI cohort with third-tertile PRS scores had earlier age at first migraine ICD code entry than males with first-tertile scores (Figure 4D,E). All Cox regression models satisfied a global test of the proportional hazards assumption, and inspection of Appendix A questions this assumption only at very long analysis times, where a relatively small number of patients developed migraine. This suggests that increased genetic susceptibility contributed to increased risk of migraine well into adult life (through age ~60, Figure 3E,G) in this population. Based on the power analyses with our sample sizes, the strength of associations with onset age (Table 3) was great enough to be detected.

In the DodoNA cohort, 43% of cases reported migraine onset during or after 2003, the first year of ICD code entry in the EHR. For this group, the median delay from AAO to first migraine ICD code was 6.8 yrs., while for patients reporting onset prior to 2003, the median delay was 28.3 yrs. (Figure 3B). This suggests that migraine ICD code may be an inconsistent proxy for onset age, which may account for the differences in the strength of association with onset age between the two cohorts, although the 95% confidence intervals for the HRs for both cohorts overlap.

### 3.4. Risk Scores Are Not Associated with Earlier Migraine Chronification

Risk-score associations with chronification in DodoNA cases were evaluated using three paradigms. In the first, where patients with chronification were compared with patients without chronification and patients without migraine, a higher PRS appeared to be associated with earlier chronification (Figure 5D). However, this likely reflects a signal from the association with age at migraine onset, as it was not seen when controls were DodoNA cases with migraine and without chronification. In that case, the association was weaker when analysis time was age, and no association was seen when the analysis time was disease duration (Figure 5D). Groups defined by risk-score tertiles did not differ in chronification (Appendix A). Since a power analysis estimated that a cohort of this size would be sufficient to detect an HR of 1.57, this cohort was underpowered to detect an association between genetic risk and chronification less than this.

## 4. Discussion

We analyzed the association of migraine genetic risk with migraine onset and chronification in two large, real-world, community-based cohorts. Our results add to the emerging understanding of how quantitation of genetic susceptibility for migraine using a PRS should be interpreted. Higher risk score distributions in patients with migraine were expected based on the criteria for cases in the GWASs used to develop the risk scores [17,22,41]. We also demonstrate that increased genetic risk is associated with earlier onset of migraine. Since the proportional hazards assumption in our models was generally satisfied except at very long analysis times, our results indicate that increased genetic susceptibility contributes to greater risk of migraine onset well into adult life, up to age 60 in this population. Our results extend the findings of an earlier report showing that higher mean PRS scores correlate with earlier age of onset in age groups under 20 years old [16] and suggest that this risk remains into later adulthood.

In the treated DodoNA cohort, the PRS was similar in migraine cases with and without chronification when chronification was assessed throughout the disease course, but was not strongly associated with earlier chronification, i.e., it did not surpass an HR of 1.57, which we would have been able to detect. This is consistent with earlier studies that used a different analytical (logistic regression) approach [35], suggesting that chronification is caused by environmental rather than genetic factors.

Considering that migraine is highly prevalent and strongly influenced by multiple factors [24,26,42], our results suggest that while higher genetic risk scores capture increased susceptibility to migraine, the PRS quantifies a genetically based susceptibility that is distinct from factors that influence disease course.

It will be important to further assess genetic-risk score associations with migraine characteristics and disease outcomes in the context of treatment, ancestry, and environmental factors. The PRS we used, like many risk scores for complex traits, has relatively low predictive power [16,43,44]; nonetheless, this work suggests that integrative genetic risk scores will be of use in stratifying patients based on susceptibility, although genetic risk may not be sufficient to predict all disease outcomes.

### Limitations

Patient self-reporting of migraine onset may be subject to recall bias. However, the association of higher genetic risk with earlier onset appears to be robust, since similar results were found using two different methods for estimating onset and also when onset was treated as a categorical variable. In addition, a practice setting excludes persons with migraine who do not seek medical attention.

This study was designed as a real-world, retrospective clinical–genetic investigation; thus, data collection was limited by what was available from study patients who had migraine-risk variant genotypes with high call rates. Including only individuals with evaluable data increases the likelihood of selection bias; however, as missing data were likely to be missing at random, this subset of patients was likely representative of the population.

While our study was hypothesis driven, the analyses were data driven and exploratory, as the variables used to specify and estimate our models were based on our data. Our results should be interpreted in the context of our patient population and analysis approach; our models may be population-specific. The PRS was developed and evaluated in individuals of European ancestry; thus, we limited our analyses to patients with this ancestry, as PRSs developed in individuals of one ancestry may not capture relevant genetic variation in other ancestries. These findings should thus be evaluated in patients with different ancestries using PRSs developed for those ancestries or with demonstrated validity across multiple ancestries.

Since this work focused on whether genetic risk is associated with the age at onset of migraine and chronification of migraine, we did not evaluate associations with non-migraine headache or migraine subtypes or characteristics (e.g., migraine with aura, episodic migraine, high-frequency episodic migraine), some of which have been previously described [16]. It will be useful to evaluate whether genetic risk is associated with other classification schema, such as disabling migraine (>7 MDM, prolonged attacks, poor symptomatic treatment, and suboptimal preventative response).

## Figures and Tables

**Figure 1 jcm-13-06483-f001:**
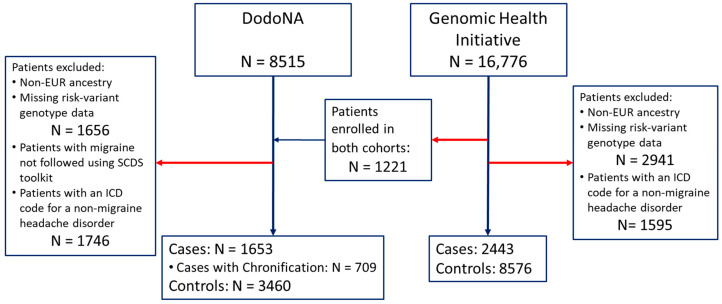
Study participants. A total of 16,776 patients enrolled in the Genomic Health Initiative (GHI) cohort were genotyped. Patients with non-EUR ancestry or not satisfying genotyping quality control were excluded (N = 2941). Patients with a non-migraine headache ICD code were excluded as controls (N = 1595). Patients concurrently enrolled in the DodoNA cohort were evaluated only in DodoNA analyses (N = 1221). This resulted in 2443 cases and 8576 controls in the GHI cohort. A total of 8515 patients enrolled in the DodoNA cohort were genotyped. Patients with non-EUR ancestry or not satisfying genotyping quality control were excluded (N = 1656). Patients with a diagnosis of migraine who were not followed using the migraine structured clinical documentation support toolkit were excluded (as cases), as were patients with an ICD code for a non-migraine headache disorder (as controls) (N = 1746). This resulted in 1653 cases and 3460 controls in the DodoNA cohort. Among DodoNA cases, 709 developed chronification of migraine during the disease course, prior to study enrollment or during follow-up. Red arrows indicate patients excluded from the study group. Blue horizontal arrow indicates patients enrolled in both cohorts who were included in the DodoNA cohort only. Blue vertical arrows indicate final numbers of patients included in study cohorts.

**Figure 2 jcm-13-06483-f002:**
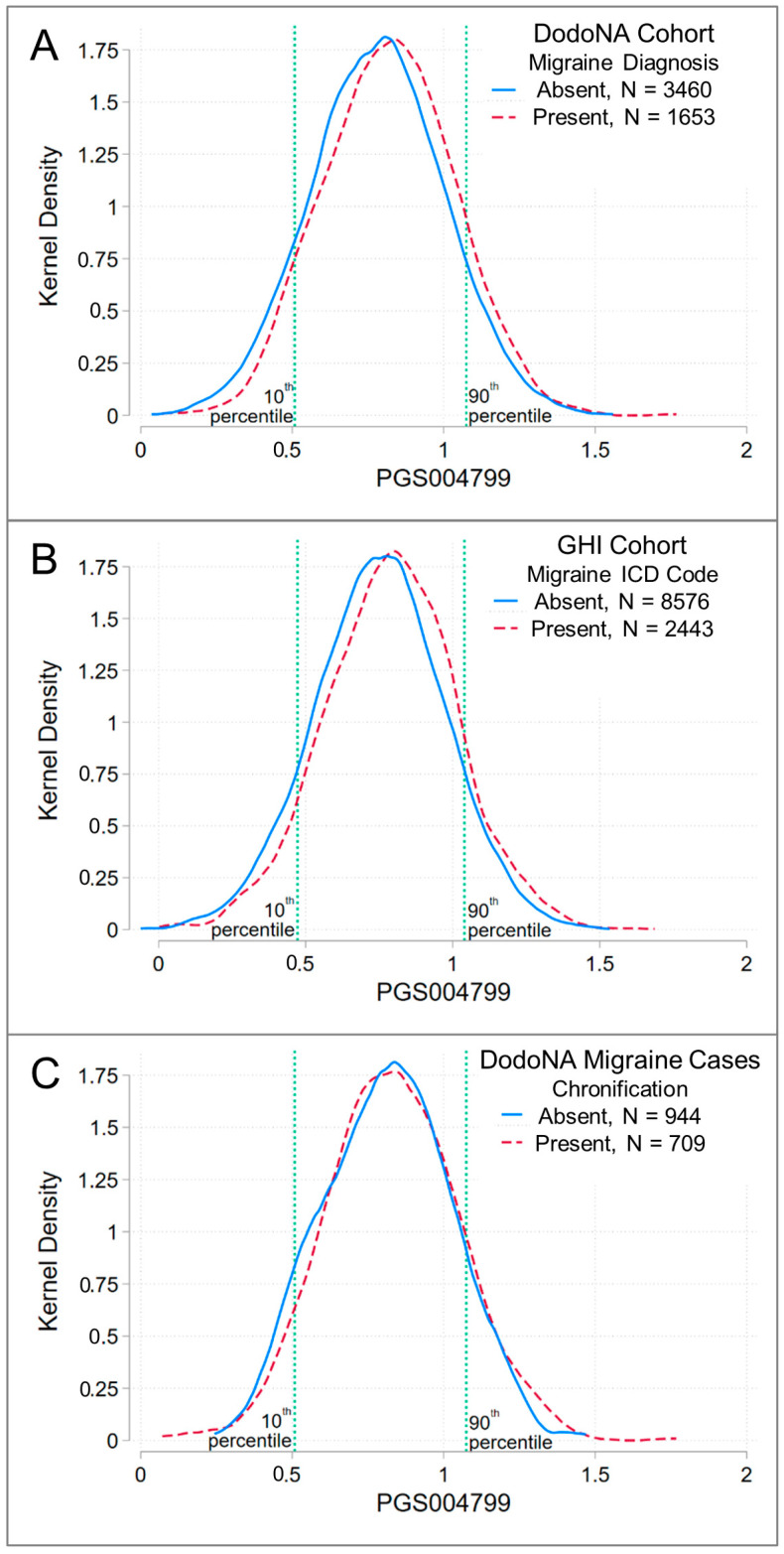
Migraine risk-score distributions in the DodoNA and GHI cohorts. Patients enrolled in the DodoNA and GHI cohorts were recruited from communities in Northeastern Illinois, USA. Kernel-density plots using an Epanechnikov kernel show the distribution of PGS004799 in cohort members with European ancestry. Panel (**A**) compares the risk-score distribution of patients enrolled in the DodoNA cohort with neurologist-diagnosed migraine (cases; red dashed line) to that in patients lacking an ICD code for a headache disorder (ICD9 346-, 339-, 307.81, 784.0; ICD10 G43-, G44-) (controls; blue solid line). Panel (**B**) shows risk-score distributions in patients from the GHI cohort. Patients with an ICD code for migraine (ICD9 346-, ICD10 G43-) (cases; red dashed line) are compared to patients without an ICD code for a headache disorder (controls; blue solid line). Panel (**C**) compares the risk-score distributions in DodoNA cohort cases with (red dashed line) and without (blue solid line) chronification of migraine. Tests of mean differences (two-tailed *t*-test) are reported. Compare to Table 2.

**Figure 3 jcm-13-06483-f003:**
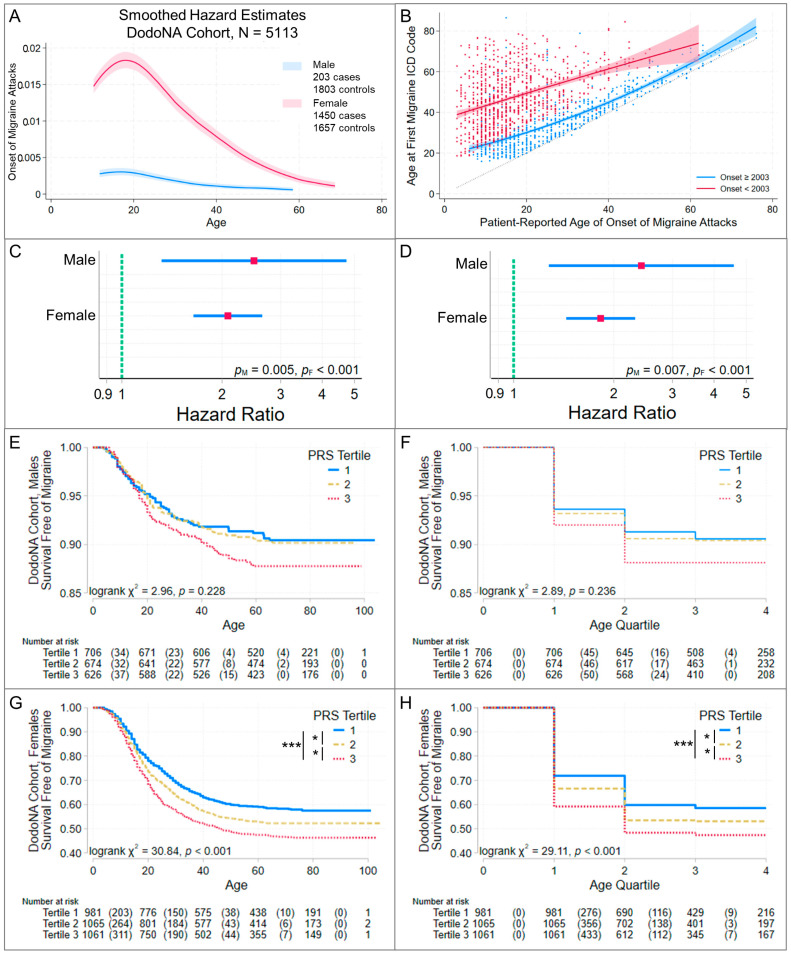
Migraine risk-score association with age of onset of migraine attacks in the DodoNA cohort. Cox proportional hazards regression was used to evaluate the association of PGS004799 with age at onset of migraine in DodoNA cohort cases—patients were enrolled with neurologist-diagnosed migraine and age at onset was ascertained at study enrollment. Controls had no ICD code for a headache disorder (ICD9 346-, 339-, 307.81, 784.0; ICD10 G43-, G44-) and were censored at the age of their last known clinical encounter. Panel (**A**) shows the number of cases and controls and smoothed hazard estimates obtained using an Epanechnikov kernel function, with the 95% pointwise confidence band shaded. Since the hazard estimates differed by sex, risk-score associations were evaluated separately for males and females. Panel (**B**) illustrates the relationship between age at onset in the DodoNA cases and age at first ICD code for migraine before (red) and after (blue) 2003, when electronic health record data became available. This informs interpretation of risk-score associations with the age at first ICD code, which was assessed in the GHI cohort (Figure 3). Quadratic-fit lines (95% confidence interval shaded) are shown. The results of Cox proportional hazard models are summarized in panels (**C**,**D**). Since patient-reported onset age may have been more reliable in younger patients and test–retest reliability was unknown, we separately evaluated age (**C**) and age quartiles (**D**) as the analysis time. The red square identifies the hazard ratio and the blue line its 95% confidence interval. Significance (*p*_M_ male, *p*_F_ female) is noted. In both sexes, a higher PRS was associated with earlier onset age. Additional analyses evaluated whether Kaplan–Meier estimates differed in patients grouped by risk-score tertiles in males (**E**,**F**) and females (**G**,**H**) using age (**E**,**G**) or age quartile (**F**,**H**) as the analysis time. If an overall log-rank test was significant, groups showing significant differences were identified using pairwise log-rank tests, Bonferroni corrected for multiple tests: *** *p* < 0.001, * *p* < 0.05. The table under each graph lists the number of patients at risk and, in parentheses, the number of failures. Compare to Table 3.

**Figure 4 jcm-13-06483-f004:**
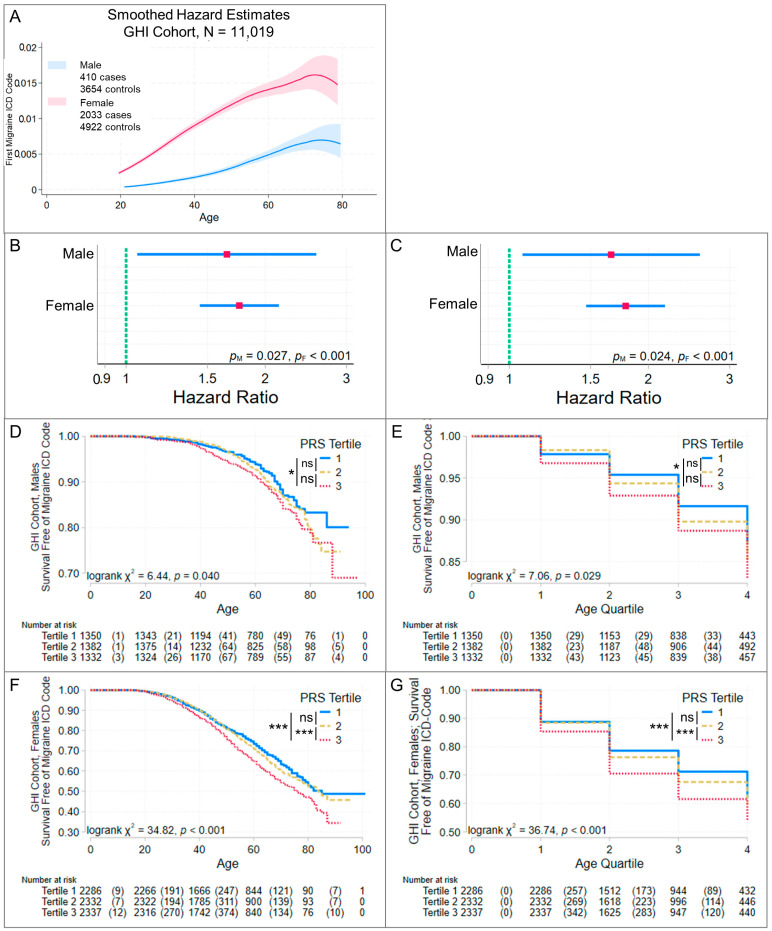
Migraine risk-score association with age of first ICD code for migraine in the GHI cohort. Cox proportional hazards regression was used to evaluate the association of PGS004799 with age at first ICD code for migraine in the GHI cohort. Cases were patients with an ICD code for migraine (ICD9 346-; ICD10 G43-); controls were patients without an ICD code for a headache disorder (ICD9 346-, 339-, 307.81, 784.0; ICD10 G43-, G44-) and were censored at the age of their last known clinical encounter. Panel (**A**) shows the number of cases and controls and smoothed hazard estimates obtained using an Epanechnikov kernel function, with the 95% pointwise confidence band shaded. Since the hazard estimates differed by sex, risk-score associations were evaluated separately for males and females. The results of Cox proportional hazard models are summarized in panels (**B**,**C**). Since first migraine ICD code may not have been a reliable estimate of onset age, we separately evaluated age (**B**) and age quartiles (**C**) as the analysis time. The red square identifies the hazard ratio and the blue line its 95% confidence interval. Significance (*p*_M_ male, *p*_F_ female) is noted. In both sexes, a higher PRS was associated with earlier onset age. Additional analyses evaluated whether Kaplan–Meier estimates differed in patients grouped by risk-score tertiles in males (**D,E**) and females (**F**,**G**) using age (**D**,**F**) or age quartile (**E**,**G**) as the analysis time. If an overall log-rank test was significant, groups showing significant differences were identified using pairwise log-rank tests, Bonferroni corrected for multiple tests: *** *p* < 0.001, * *p* < 0.05, ns = not significant. The table under each graph lists the number of patients at risk and, in parentheses, the number of failures. Compare to Table 3.

**Figure 5 jcm-13-06483-f005:**
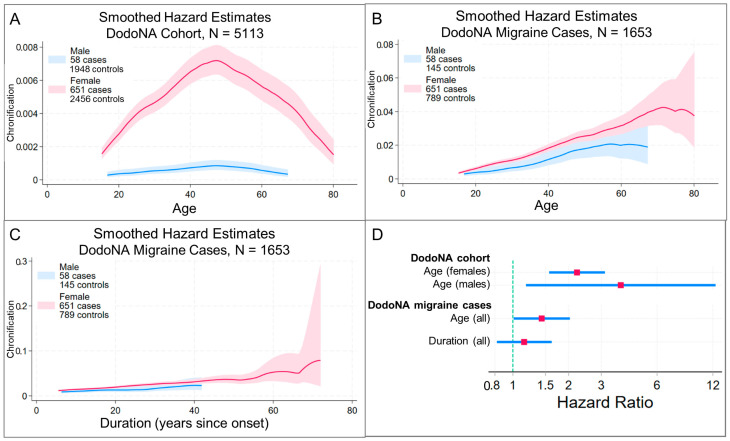
Migraine risk-score association with chronification in the DodoNA cohort. Cox proportional hazards regression was used to evaluate the association of PGS004799 with chronification of migraine in DodoNA cases. Three paradigms were assessed. Each defined cases as patients who developed chronification prior to the last follow-up visit and used neurologist-documented age at first chronification. Panels (**A**–**C**) show the number of cases and controls used in each paradigm and the smoothed hazard estimates obtained using an Epanechnikov kernel function, with the 95% pointwise confidence band shaded. In the first paradigm (**A**), controls were DodoNA patients without migraine and DodoNA cases without chronification; analysis time was age. Since the hazard estimates differed by sex in this paradigm, risk-score associations were evaluated separately for males and females. In the second paradigm (**B**), controls were DodoNA cases without chronification and analysis time was age. In the third paradigm (**C**), controls were DodoNA cases without chronification and analysis time was the number of years from migraine onset. Controls were always censored at the age of their last known clinical encounter. Panel (**D**) summarizes the results. The red square identifies the hazard ratio and the blue line its 95% confidence interval. Results using the first paradigm suggested that a higher PRS was associated with earlier chronification in both sexes. However, this likely reflects the signal from the association with age at migraine onset, as the association with chronification was less strong when controls were DodoNA cases with migraine but without chronification and analysis time was age, and no association was seen when controls were DodoNA cases with migraine but without chronification and analysis time was disease duration. Compare to Table 3.

**Table 1 jcm-13-06483-t001:** Characteristics of study participants. ^a^

A. DodoNA Cohort			
**Characteristic**	**Cases**	**Controls ^b^**	**Total**
N (% of cohort)	1653 (32.3)	3460 (67.7)	5113 (100)
Female, N (%)	1450 (87.7)	1657 (53.3)	3107 (100)
Age at study enrollment, median (range)	42 (18–86)	68 (18–99)	60 (18–99)
Age at last clinical encounter, median (range)	48 (19–93)	73 (19–104)	65 (19–104)
Patient reported age at onset, median (range)	18 (3–76)	−	−
Disease duration (years), median (range)	18.3 (1–72)	−	−
Education years, N (%)		−	−
<12	4 (0.2)		
12	152 (9.2)		
≤16	802 (48.5)		
>16	636 (38.4)		
Not reported	59 (3.6)		
BMI at baseline, N (%)			
<18.5	48 (2.9)		
18.5–24.9	720 (43.6)		
25–29.9	475 (28.7)		
30–34.9	211 (12.8)		
35–39.9	100 (6.0)		
>40	86 (5.2)		
Not reported	13 (0.8)		
Migraine with aura at enrollment, N (%)	659 (39.9)	−	−
Chronification of migraine, N (% of cases) (over all follow-ups)	709 (42.9)	−	−
Follow up years, mean	2.29		
Median (range)	2 (1–10)		
Interquartile range	1–3		
Age at chronification, median (range)			
All (N = 709)	42 (10–85)		
Males (N = 58)	42 (10–74)		
Females (N = 651)	42 (10–85)		
Years from migraine onset to chronification, all (N = 709)	19 (0.2–77)		
Males (N = 58)	19 (0.2–48)		
Females (N = 651)	19 (0.2–77)		
Migraine Disability Assessment (MIDAS) Part A at enrollment, median (range)	21 (1–90), N = 1534	−	−
Migraine Disability Assessment (MIDAS) Part B at enrollment, median (range)	7 (2–10), N = 1587	−	−
Migraine-Specific Quality of Life (MSQ) at enrollment, median (range)	58 (4–84), N = 1612	−	−
B. GHI Cohort			
**Characteristic**	**Cases ^c^**	**Controls ^b^**	**Total**
N (% of cohort)	2443 (22.2)	8576 (77.8)	11,019 (100)
Female, N (%)	2033 (83.2)	5922 (57.4)	6955 (63.1)
Age at study enrollment, median (range)	54 (18–91)	60 (18–101)	59 (18–101)
Age at first migraine ICD code, median (range)	48 (11–88)	−	−

^a^ See Figure 1 and the text for a description of patient inclusion and exclusion criteria. ^b^ Absence of ICD code for a headache disorder: ICD9 codes 346-, 339-, 307.81, 784.0 and ICD10 codes G43- and G44-; ^c^ presence of an ICD code for migraine: ICD9 code 346- or ICD10 code G43-.

**Table 2 jcm-13-06483-t002:** PGS004799 scores in cases and controls of the DodoNA and GHI cohorts.

Participants	N	PGS004799		
		Mean (SD)	Median (Range)	Evaluation of Differences ^a^
DodoNA cohort	5113	0.794 (0.219)	0.7935 (0.036–1.767)	−
Cases, neurologist-diagnosed migraine	1653	0.821 (0.215)	0.822 (0.072–1.767)	*t* (5111) = −6.06, *p* = 1.42 × 10^−9^ KS: *D* = 0.077, *p* = 3.88 × 10^−6^
Controls	3460	0.781 (0.220)	0.782 (0.036–1.559)	
Cases with chronification	709	0.831 (0.220)	0.827 (0.072–1.767)	*t* (1651) = −1.67, *p* = 0.094 KS: *D* = 0.0423, *p* = 0.463
Cases without chronification	944	0.813 (0.210)	0.816 (0.243–1.469)	
Genomic Health Initiative (GHI) Cohort	11,019	0.759 (0.224)	0.763 (−0.061–1.687)	−
Cases	2443	0.790 (0.223)	0.792 (0.003–1.687)	*t* (11,017) = −7.69, *p* = 1.58 × 10^−14^ KS: *D* = 0.0792, *p* = 8.64 × 10^−11^
Controls	8576	0.750 (0.223)	0.754 (−0.060–1.534)	

^a^ Mean differences evaluated using a two-tailed *t*-test; equality of distribution functions evaluated using a Kolmogorov–Smirnov (KS) test.

**Table 3 jcm-13-06483-t003:** Tests of association of PRS with age of onset of migraine attacks and chronification of migraine.

A. Cox proportional hazards models																			
**Population**		**Diagnosis**		**Diagnostic Criteria**					**Analysis Time**		**Sex**		**Cases/Total**		**HR** **(95% CI)**		**z**		** ^a^ *p* **
				**Cases**		**Controls**													
DodoNA Cohort		Migraine		ICHD-3		Absence of ICD9 codes 346-, 339-, 307.81, 784.0 and ICD10 codes G43-, G44-			Age		Female		1450/3107		2.08 (1.64–2.64)		6.05		<**0.001**
											Male		203/2006		2.49 (1.32–4.73)		2.80		**0.005**
									Age quartiles		Female		1450/3107		1.83 (1.44–2.32)		4.95		<**0.001**
											Male		203/2006		2.42 (1.28–4.59)		2.71		**0.007**
		Chronification of migraine		ICHD-3		ICHD-3 in DodoNA case, otherwise absence of ICD9 codes 346-, 339-, 307.81, 784.0 and ICD10 codes G43-, G44-			Age		Female		651/3107		2.22 (1.57–3.14)		4.51		<**0.001**
											Male		58/2006		3.82 (1.18–12.41)		2.23		**0.026**
DodoNA cases		Chronification of migraine		ICHD-3		ICHD-3			Age		Both		709/1653		1.43 (1.01–2.03)		2.04		**0.041**
									Years from onset		Both		709/1653		1.15 (0.82–1.62)		0.80		0.424
GHI cohort		Migraine		Presence of ICD9 code 346- or ICD10 code G43-		Absence of ICD9 codes 346-, 339-, 307.81, 784.0 and ICD10 codes G43-, G44-			Age		Female		2033/6955		1.76 (1.44–2.14)		6.34		<**0.001**
											Male		410/4064		1.65 (1.06–2.58)		2.21		**0.027**
									Age quartiles		Female		2033/6955		1.79 (1.47–2.17)		5.80		<**0.001**
											Male		410/4064		1.66 (1.07–2.59)		2.25		**0.024**
B. Log-rank tests of equality of survivor functions for age at onset in groups defined by risk-score tertiles																			
**Cohort**	**Diagnosis**		**Diagnostic Criteria**				**Analysis Time**	**Sex**		**Cases/** **Total**		**aLog-Rank Test χ2, *p***							
			**Cases**		**Controls**							**Overall**		**Tertile** **1 vs. 2**		**Tertile** **1 vs. 3**		**Tertile** **2 vs. 3**	
DodoNA cohort	Migraine		ICHD-3		Absence of ICD9 codes 346-, 339-, 307.81, 784.0 and ICD10 codes G43-, G44-		Age	Female		1450 /3107		29.11, <**0.001**		7.11, **0.023**		28.88, <**0.001**		7.81, **0.016**	
								Male		203 /2006		2.96, 0.228		−		−		−	
							Age quartiles	Female		1450 /3107		30.84, **0.001**		7.39, **0.020**		30.67, <**0.001**		8.36, **0.011**	
								Male		203 /2006		2.89, 0.236		−		−		−	
	Chronification of migraine		ICHD-3		ICHD-3 in DodoNA cases, otherwise absence of ICD9 codes 346-, 339-, 307.81, 784.0 and ICD10 codes G43-, G44-		Age	Female		651 /3107		18.84, **0.001**		3.78, 0.155		18.64, <**0.001**		5.80, **0.048**	
								Male		58 /2006		2.58, 0.275		−		−		−	
DodoNA cases	Chronification of migraine		ICHD-3		ICHD-3		Age	Both		709 /1653		4.77, 0.092		−		−		−	
							Years from onset	Both		709 /1653		1.64, 0.441		−		−		−	
GHI cohort	Migraine		Presence of ICD9 code 346- or ICD10 code G43-		Absence of ICD9 codes 346-, 339-, 307.81, 784.0 and ICD10 codes G43-, G44-		Age	Female		2033 /6955		34.82, <**0.001**		1.77, 0.183		30.51, <**0.001**		18.35, <**0.001**	
								Male		410 /4064		6.44, **0.040**		1.56, 0.636		6.56, **0.031**		1.68, 0.583	
							Age quartiles	Female		2033 /6955		36.74, <**0.001**		3.01, 0.249		33.71, <**0.001**		17.14, <**0.001**	
								Male		410 /4064		7.06, **0.029**		1.92, 0.498		7.04, **0.024**		1.70, 0.576	

^a^ Between-group *p*-values are Bonferroni corrected for multiple tests. *p* < 0.05 in bold font. Abbreviations: HR, hazard ratio; CI, confidence interval; ICHD-3, International Classification of Headache Disorders 3rd edition.

## Data Availability

Summary statistics are available from the corresponding author upon reasonable request, following institutional approval.

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
