# Peer review of "An Integrative Migraine Polygenic Risk Score Is Associated with Age at Onset But Not Chronification"

_jcm, 2024, doi:10.3390/jcm13216483_

Round 1

Reviewer 1 Report

Comments and Suggestions for Authors

The authors have carried out an interesting , well-designed and well-conducted study in which they provide data, consistent with previous works, on the genetic susceptibility of migraine, but not its chronification.

It seems clear that the chornic migraine situation as actually defined cannot be predicted genetically. However, and I would like to see it reflected in the discussion, a new concept of disabling migraine (more than 7 days per month of migraine, prolonged attacks with difficult response to specific symptomatic treatments, migraine with suboptimal response to preventives) could, if measured, be genetically predicted. 

Author Response

We thank Reviewer 1 for their comments and insightful suggestion.  We have modified the discussion to address this issue, and an issue raised by Reviewer 2, on lines 412-418. 

Reviewer 2 Report

Comments and Suggestions for Authors

This is a good study, but we are unsure how you’re using genetic information to confirm risk and how this relates to questionnaire.

Specifically, it is difficult to understand the relationship between Migraine polygenic risk scores (PRS)  and the GWAS data.

Is there a new methodology of scoring your developing? are you developing a new tool for migraine evaluation? are you are you evaluating?This cccc

Comments on the Quality of English Language

Is this a new tool and where is the validity data?

when structuring sentences try to indicate clearly the relationship you are trying to prove.

The study seems broad, but lack specific information as to how you came to the conclusion of using the suggest tool.

The study just needs some more clarification of the relationship above listed and your conclusion.

also state limitations such as headache and migraine subtypes were not done.

a good effort 

Author Response

Response:  We thank Reviewer 2 for their valuable comments.

Regarding the relationship between the migraine PRS and GWAS data, we describe how a PRS is developed (in general) in the Introduction on lines 63-65, and we describe some details of the development of the PRS used in this study in the Methods section (lines 154-159).  While a more detailed description of how PRS are developed would more fully address the question about the relationship between GWAS data and the PRS, this would take considerable space.  We have modified the Introduction to provide additional perspective on the utility of a PRS and added a citation for a classic paper by Inouye et al. (2018) that provided compelling evidence for how to design a PRS with utility for risk prediction in a clinical setting.  Since we did not develop or validate the PRS we used, we cite the authors that did this work (Truong et al. 2024) and describe in some detail how we calculated the PRS that they validated (lines 159-162).  We feel that this information is sufficient explanation for this manuscript, as identifying PRS with clinical utility has been an active area of research since 2018, when Inouye et al. showed that a PRS could have a similar predictive ability of a monogenic mutation (for cardiovascular disease) and strategies to develop PRS based on GWAS have evolved considerably since this time. 

Regarding the toolkit that we used to collect patient data, we have added text to clarify that the toolkit was used only as a vehicle to standardize the collection of patient data across neurologists and their patients over time, and not a new tool per se (Methods, lines 90-93).  We used the ICHD-3 guidelines for diagnosis, so did not use or develop a new tool for migraine characterization.  The toolkit has been described previously (citations 35-36) and was not intended to be a tool for diagnosis or characterization.  Table S1 describes the data elements captured using the tool, which include information that provided confidence in an ICHD-3-based diagnosis of migraine (and non-migraine) headaches, details of migraine characteristics and symptoms, medication responses, scores on previously validated measures of migraine disability, frequency and severity (MSQ, MIDAS-A, MIDAS-B), and scores on previously validated assessments of migraine-related comorbidities (ISI, CES-D, GAD-7).  Collection of data on the age at first migraine symptom and first diagnosis, and prior medical history related to chronification were directly pertinent to this research .

We understand that the study seems broad and that we carried out several analyses.  However, the study focused on two main questions: (1) Is genetic risk related to age of migraine onset and (2) Is genetic risk associated with chronification of migraine.  We carried out analyses to demonstrate that the associations we found we robust and did not have alternate explanations. For example, since patient-reported age of onset may be biased for a number of reasons, we evaluated associations using patient reported age of onset as well as age-quartile as analysis time.  Both showed similar associations.  As another example, since first migraine-related ICD code was used as a proxy for first diagnosis in the GHI cohort, we assessed the relationship between age at first ICD code and patient-reported age at onset.

While these additional analyses make the paper more complex, they also add to its rigor. While the figures and tables take considerable space, the results are summarized in just over 60 lines of text that is organized into paragraphs with headers to guide the reader, and the first two paragraphs of the discussion summarize the main conclusions.  

To address the final comment that we did not assess associations with other types of headache or subtypes of migraine (and also address a comment from Reviewer 1), we have added a paragraph on lines 413-419 of the Discussion

Reviewer 3 Report

Comments and Suggestions for Authors

The study was conducted with scientific rigor also considering the complex topic of the genotype today much studied, but not always correct. It is original both in the approach and in the demonstration of the results that were conducted in a detailed way. The originality of the study concerns the lack of correlation between genotypic set and chronicity of headache. This data is partly in contrast with the literature produced so far, but represents a good result for demonstrating how important the environmental component is for the chronicity of headache. The bibliography reported is particularly rich and adherent to the topic studied.

Author Response

Response:  We thank Reviewer 3 for their comments.  We agree that our study provides insight into the interaction between genetic and environmental risk in chronicity of migraine; however, as we state in the first two paragraphs of the discussion, our results are generally consistent with, and extend previous work. 

Reviewer 4 Report

Comments and Suggestions for Authors

Well done study and well written with professional scientific conduct. I suggest including the limitations of solely including Caucasian and Southeast Asian races in the GWAS study dataset, which cannot represent other parts of the world. 

Author Response

Response:  We thank Reviewer 4 for their insightful comment.  The PRS we used, PGS004799, was developed and then evaluated only in European populations (lines 156-161, see also pgscatalog.org/score/PGS004799).  The PRSmixPlus pipeline was also used to develop a separate PRS, PGS004800, based on GWAS in a (British) cohort with South Asian ancestry and evaluated in a population with South Asian ancestry.  We did not evaluate PGS004800 because of an insufficient number of individuals with this ancestry in of our patient population. We fully agree that not evaluating other ancestries using a PRS developed for those ancestries or a PRS that has been validated across multiple ancestries is a limitation of our study.  We address this issue in the Discussion, lines 406-411.

Reviewer 5 Report

Comments and Suggestions for Authors

Dear Authors, thank you for your very interesting article. Unfortunately, the number of figures and tables makes it difficult to read easily. I would like to ask for them to be reduced. Also, results should be kept to a minimum. When reading them, the reader is discouraged by the amount of text.

Author Response

Response:  We appreciate Reviewer 5’s comments and thank the reviewer.  We would like to echo part of our response to Reviewer 2 as to why the study feels broad and argue that the number of display items (3 Tables, 5 Figures) is reasonable.  Including these items allows the reader to understand how we selected the study population, understand how the PRS differs in subgroups, and critically evaluate our results.  More specifically:

  • Table 1 presents demographics, which are important to describe the cohorts we studied, and are reasonably standard to include as a Table within the main body of a scientific paper.
  • Figure 1 presents an overview of how many patients were screened, excluded and included, and why. Our sense is that this is essential for the reader to understand the nature of the cohorts we studied, their differences, and evaluate potential biases arising from studying these populations in a retrospective manner.
  • Table 2 and Figure 2 present descriptive statistics and information on the distribution of PGS004799 in the two cohorts we studied. This documents whether or not the PRS differs in these groups, so sets a framework for the Cox proportional hazards and Kaplan-Meier analyses to understand whether onset and chronification differ by age at migraine onset, and so are fundamental to include. While they could be included in Supplemental Material instead of in the main text, our experience is that most readers do not examine the Supplemental Material.  Presenting this information in graphical and tabular from allows the reader to evaluate the PRS and the quality of our data.
  • Table 3 presents the results of the Cox proportional hazards models for PRS and PRS-tertiles using age and age-quartiles as analysis time, and presents the results of sensitivity analyses to address concerns about the imprecision of patient recall of onset age. Since these are the main conclusions of the manuscript, this information is essential to include.
  • Figures 3-5 summarize the associations we see with the PRS (Figure 3: DodoNA cohort, association with age at onset; Figure 4: GHI cohort: association with age at onset; Figure 5: DodoNA cohort, association with AAO). We feel these tables summarize the data clearly and are effective at showing the results.  Since they include details that are highly pertinent to our main conclusions, we argue that they should be retained. 
  • We did place material that the typical reader would not review in the Supplemental Material (details of the structured clinical documentation support toolkit used to collect our data, evaluation of the proportional hazards assumptions for our models). We feel that the display items that are in the body of the manuscript should not be relegated to this material, as they allow the reader to evaluate our data and draw their own conclusions.

The Results section is relatively short, about 60 lines.  We believe it is quite focused and effectively guides the reader through the tables and figures.  

Regarding the volume of text, we suspect this concern at least partly reflects the previous formatting of the figure legends.  In the version of the submission we downloaded for revision, the figure legends and text are interspersed and have the same sized font, while in published JCM articles, the figure legends have a smaller font and are clearly associated with a figure.  The inability to readily distinguish figure legends and text makes the manuscript difficult to read and certainly confusing. We have formatted the revised version correctly to make the distinction between text and figure legends clearer.  Also, we have reduced the size of the figures (the editor can re-adjust their size if desired) and re-organized the panels of Figure 5 so the figures align with the text and each figure is positioned near its legend. More specifically,

  • The initial text of the Methods is now on lines79-96 and 104-108, while the legend for Figure 1 is now grouped with its title on lines 110-121 and has a smaller font than the text. Previously, the legend for Figure 1 was in the same-sized font as the main text and was not grouped with its title. If the legend is read as a continuation of the text, this would be an understandable source of confusion.
  • The Results section comprises about 60 lines of text (lines 212-231, 250-268, 322-340). As was the case for the legend to Figure 1, the legends for Figures 2-5 were not clearly demarcated from this text. The legends for Figures 2-5 are now joined with their titles and use a smaller font.  This makes them more readily distinguished from the main text.